

# Divergent water sources of three dominant plant species following precipitation events in enclosed and mowing grassland steppes

Tiejun Bao[1,2,3], Yunnuan Zheng[1,2,3], Ze Zhang[1,2,3], Heyang Sun[1,2,3], Ran Chao[1,2,3], Liqing Zhao[1,2,3], Hua Qing[1,2,3], Jie Yang[1,2,3] and Frank Yonghong Li[1,2,3]

[1] Inner Mongolia University, School of Ecology and Environment, Hohhot, China
[2] Inner Monoglia University, Ministry of Education Key Laboratory of Ecology and Resource Use of the Mongolian Plateau, Hohhot, China
[3] Inner Monoglia University, Inner Mongolia Key Laboratory of Grassland Ecology, School of Ecology and Environment, Hohhot, China

Corresponding author
Jie Yang, jyang@imu.edu.cn

## ABSTRACT

Understanding of the dynamic patterns of plant water use in a changing environment is one of foci in plant ecology, and can provide basis for the development of best practice in restoration and protection of ecosystem. We studied the water use sources of three coexisting dominant plant species *Leymus chinensis, Stipa grandis* and *Cleistogenes squarrosa* growing in both enclosed and mowing grassland in a typical steppe. The oxygen stable isotope ratios ($\delta^{18}O$) of soil water and stem water of these three species were determined, along with soil moisture, before and after precipitation events. The results showed that (1) mowing had no significant effect on the soil moisture and its $\delta^{18}O$, whereas precipitation significantly changed the soil moisture though no significant effect detected on its $\delta^{18}O$. (2) *C. squarrosa* took up water majorly from top soil layer due to its shaollow root system; *L. chinensis* took up relative more water from deep soil layer, and *S. grandis* took up water from the middle to deep soil layers. (3) *L. chinensis* and *S. grandis* in mowing grassland tended to take up more water from the upper soil layers following precipitation events, but showed no sensitive change in water source from soil profile following the precipitation in the enclosed grassland, indicating a more sensitive change of soil water sources for the two species in mowing than enclosed grassland. The differences in root morphology and precipitation distribution may partly explain the differences in their water uptake from different soil layers. Our results have important theoretical values for understanding the water competition among plants in fluctuating environment and under different land use in the typical steppe.

## INTRODUCTION

Co-existing plants may use different water sources, which is one of key mechanisms in plant community construction (*Zhang et al., 2014*). The main water sources for plant utilization are precipitation, soil water, runoff water and groundwater (*Duan et al., 2007*). The proportion of various water sources in plant water uptake depends on many factors,

such as topography, land use type, soil texture and the pattern, intensity and frequency of precipitation. For example, it was observed that a single strong precipitation event could increase the availability of the water in deep soil layers, which in turn facilitated the growth of deep-rooted plant species (*Nippert & Knapp, 2007*; *Goldstein & Suding, 2014*). Plants have varied utilization efficiencies for various water sources in different ecosystems. In the desert ecosystem of southern Utah, USA, *Ehleringer et al. (1991)* found that the growth of annual and perennial succulent plants depended entirely on summer precipitation, whereas herbaceous and perennial woody plants could use both summer and winter-spring precipitation, with herbaceous plants having more dependence on summer precipitation. Analyzing plant water sources helps understanding of the adaptation mechanisms used by different species to cope with the arid environment, thus providing a basis for accurately addressing the root water uptake when constructing hydrological models. In addition, these analyses can also, according to the spatial and temporal differences in plant water sources, provide guidance for species selection and matching in revegetation to avoid excessive competition among species. Currently, the studies about the use of water sources by plants have been widely conducted in deserts, temperate forests (*Halliday, 2011*), Mediterranean-type deserts (*Matimati et al., 2013*) and coasts (*McCole & Stern, 2007*; *Corbin et al., 2005*).

The temperate semi-arid steppe is one of the most important ecosystems in the world, covers approximately one-fifth of land surface, and the typical steppe covers 10.5% of the territory in China (*Wang, 2011a*). Precipitation is largely the only water source in semi-arid steppe ecosystems, so the water consumed by plants is mainly from soil water after the redistribution of precipitation in soil. For example, the deep-rooted shrub *Caragana microphylla* mainly utilizes water from deep soil layer that is derived from winter snowfall and heavy precipitation; whereas the shallow-rooted grass *Cleistogenes squarrosa* mainly utilizes surface soil water that is dependent on summer precipitation (*Yang et al., 2011*). Mowing (for hay) is one of the main utilization modes of semi-arid steppes. Mowing not only affects the redistribution of precipitation in the soil by affecting the canopy structure of the community (*Zhang et al., 2014*; *Wang et al., 2015*), but also affects the functional traits of plant roots (*Zhang et al., 2014*), and these changes would inevitably affect the water sources of dominant plant species. Compared with other ecosystems, the semi-arid steppe ecosystem is more sensitive to changes of water resources (*Lioubimtseva et al., 2005*; *Zhou, Li & Zhu, 2015*), and more responsive to transient fluctuations in resource availability (*Zhang et al., 2014*). Therefore, understanding the mechanisms of plant water use is essential to steppe management by exploring how dominant plants respond to instantaneous precipitation events and use available water resources in enclosed and mowing conditions.

Traditional methods of studying plant water sources were difficult, such as root excavation could determine the available water sources but the main water sources cannot be determined, because the existence of roots does not mean that these roots are active in water absorption (*Flanagan, Marshall & Ehleringer, 1993*). Comparatively, stable isotope technology has high sensitivity and accuracy, and has wide applications in the study of water sources and water use efficiency of plants in natural ecosystems (*Yoder & Nowak,*

*1999*; *Vandenschrick et al., 2002*; *Schwinning, Starr & Ehleringer, 2005*; *Nippert & Knapp, 2007*; *Goldstein & Suding, 2014*). In this study, we aimed to explore how the three coexisting dominant species *Stipa grandis*, *Leymus chinensis* and *Cleistogenes squarrosa* in a typical steppe community respond to summer precipitation events in water use under two grassland utilization modes (mowing and enclosed). The species are major dominants of the typical steppe in the region (*Bai et al., 2008*). We determined the oxygen stable isotope ratios ($\delta^{18}O$) of soil water and stem water of these three plants, and soil water content before and after precipitation events in the mowing and enclosure plots. The results would reveal the water use pattern and the competition relationship of plants in the typical steppe, and have important theoretical significance for understanding the relationship between plants and the environment under degradation in arid and semi-arid regions.

## MATERIALS AND METHODS

### Study area

This experiment was conducted in the Grassland Ecosystem Research Station of the Inner Mongolia University, located 40 km east of Xilinhot city in Central Inner Mongolia, China (116°2′–116°30′E, 44°48′–44°49′N, 1,101 m asl). The region experiences a temperate semi-arid climate, with a mean annual temperature of 2.6 °C, the annual accumulated temperature of 2,412 °C (>0 °C), and the average period of plant growth approximately 150 days; mean annual precipitation is between 200 and 350 mm, 78% of which falls between June and September, and the annual evaporation is 1,600–1,800 mm (*Wan et al., 2016*; *Bai et al., 2018*). The major soil type is a sandy loam chestnut soil, equivalent to Calcic-orthic Aridisol in the US soil taxonomy classification system.

The vegetations in the study region are largely dominated by *S. grandis*, with *L. chinense* and *Cleistogenes squarrosa* as major species. The enclosure and mowing plots were started in 2011 to study the effects of annual mowing on native steppe. Eight 20 × 30 m experimental plots (four enclosure and four mowing plots) were established with a distance of five m between any two plots. The mowing treatment was performed on August 20th of each year since 2011.

### Sample collection

Two precipitation events were recorded in July 2016, a light precipitation event of 10.8 mm on 29 July and a medium precipitation event of 20.0 mm on 30 July. Samples were collected separately on the day before these precipitation events (28th July) and on the first and fifth day after precipitation (31st July and 4th August) from each experimental plot. Non-photosynthetic tissues of plant from the interface between shoot and root systems were collected for the analysis of oxygen stable isotope ratios (*Thorburn & Walker, 1993*). For each plant species, the non-photosynthetic tissues from at least 20 individuals were collected in each plot and combined as one replicate, enclosed in the screw-capped glass vial, immediately sealed with Parafilm, and then stored at −20 °C for further stable oxygen isotope analysis.

The soil samples at the depth of 0–5, 5–10, 10–20, 20–40, 40–60, 60–80 and 80–100 cm in each plot were collected with a five cm diameter soil auger. One part of these soil samples was immediately placed into a screw-capped glass vial, sealed with parafilm and then stored at −20 °C for stable oxygen isotope analysis. The other part of the soil sample was placed into an aluminum box and weighed to obtain the fresh weight, and then weighed after oven-drying at 105 °C to get soil water content.

## Water extraction and sample analysis

The water from the soil and plant samples was extracted with a cryogenic vacuum distillation extraction system (*Ehleringer & Osmond, 2000*). The water isotope analyzer (LGR 912-0032, USA) was used to determine the $\delta^{18}O$ of the extracted water samples with a determination precision of 0.1‰. Three kinds of laboratory working standard water were measured additionally after every three samples against one of them as reference. The isotope ratio of oxygen in water is expressed by the standard delta ($\delta$) notation in parts per thousand (‰) as follows:

$$\delta^{18}O \ = \ \frac{R_{sample} - R_{standard}}{R_{standard} \times 1{,}000} \tag{1}$$

where $R_{sample}$ and $R_{standard}$ are the molar ratios of $^{18}O/^{16}O$ of the water sample and standard water (V-SMOW), respectively.

## Data analysis

Experimental data were analyzed using SPSS Version 19.0 (SPSS Inc., Chicago, IL, USA). Three-way analysis of variance (ANOVA) was conducted to test the effects of the sampling time (ST), mowing (C) and soil depth (SD) on the soil water content and $\delta^{18}O$ with all of the factors and their interactions being treated as fixed effects. The two-way ANOVA was also used to examine the effects of the ST and mowing (C) on the $\delta^{18}O$ of plant water. In addition, the differences in the $\delta^{18}O$ of plant water between the enclosure and mowing treatments were tested using the independent sample *t*-test.

Potential water sources for the three plant species were divided into four layers of soil water, which are (1) surface water layer (0–5 cm), (2) shallow water layer (5–10 cm), (3) middle water layer (10–40 cm) and (4) deep water layer (40–100 cm). An IsoSource model (*Phillips & Gregg, 2003*) was used to compare the isotope values of the water in the plant xylem and the isotope values of various potential water sources, and thereby obtaining the feasible ranges of the different water sources used by the three plant species at each ST (*Phillips & Gregg, 2003*). The source increment was defined as 2%, and the mass balance tolerance was defined as 0.01‰.

## RESULTS AND ANALYSIS

### Effects of summer precipitation on soil water content in mowing and enclosure steppe

Soil water content exhibited a very significant change before and after precipitation (Table 1; Fig. 1). As expected, the lowest soil water content appeared before precipitation

**Table 1 ANOVA result for mowing (C), sampling time (ST) and soil depth (SD) on the soil water content.**

| Source | Type III sum of squares | df | Mean square | F | P |
|---|---|---|---|---|---|
| C | 0.000 | 1 | 0.000 | 0.99 | 0.323 |
| ST | 0.002 | 2 | 0.001 | 8.756 | 0.000 |
| SD | 0.028 | 6 | 0.005 | 46.242 | 0.000 |
| C * ST | 0.000 | 2 | 5.46E-05 | 0.534 | 0.588 |
| C * SD | 0.001 | 6 | 0.000 | 1.320 | 0.257 |
| ST * SD | 0.001 | 12 | 8.68E-05 | 0.849 | 0.601 |
| C * ST * SD | 0.001 | 12 | 8.23E-05 | 0.805 | 0.645 |
| Error | 0.009 | 84 | 0.000 | | |
| Total | 0.684 | 126 | | | |
| Corrected total | 0.042 | 125 | | | |

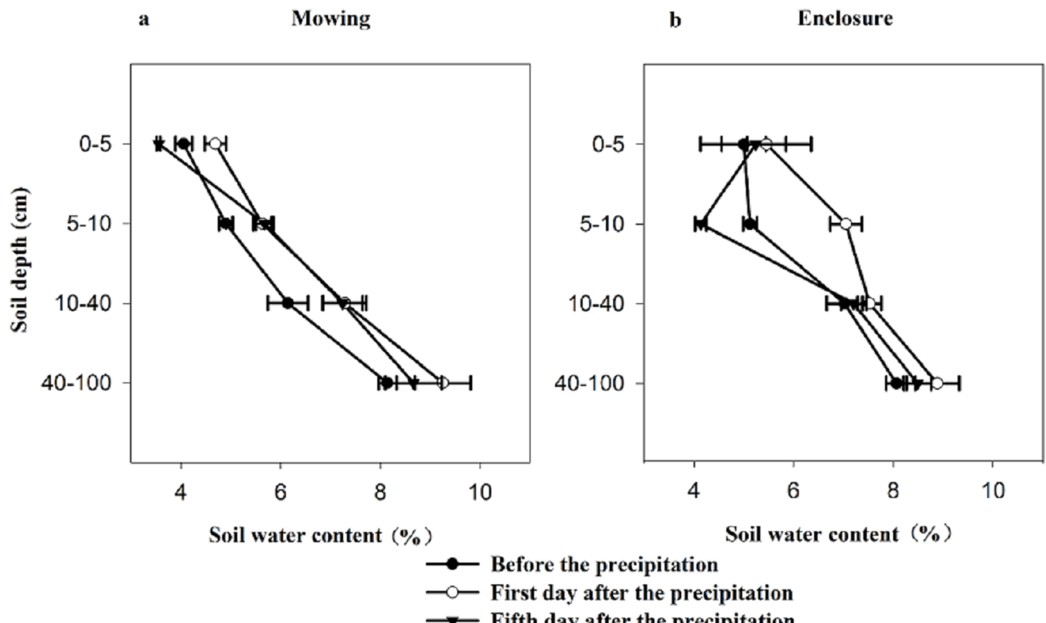

**Figure 1 Characteristics of soil water content before and after precipitation under mowing (A) and enclosure (B) treatments.**

and the highest soil water content appeared on the first day after precipitation. The changes in the soil water contents with the SD were different between the two experiment states. Under the mowing and enclosure treatment, with the increasing SD the soil water content increased, except the fifth day after the precipitation in enclosure grassland (Fig. 1).

## Effects of summer precipitation on δ¹⁸O of soil water in mowing and enclosure steppe

Both mowing and the ST had no significant effects on the $\delta^{18}O$ of soil water, but the interaction between the two factors and the SD had a significant effect on soil water $\delta^{18}O$

**Table 2 ANOVA result for mowing (C), sampling time (ST) and soil depth (SD) on soil water $\delta^{18}O$.**

| Source | Type III sum of squares | df | Mean square | F | P |
|---|---|---|---|---|---|
| C | 1.237 | 1 | 1.237 | 0.647 | 0.423 |
| ST | 2.371 | 2 | 1.185 | 0.620 | 0.540 |
| SD | 907.113 | 6 | 151.185 | 79.100 | 0.000 |
| C * ST | 15.850 | 2 | 7.925 | 4.146 | 0.018 |
| C * SD | 7.928 | 6 | 1.321 | 0.691 | 0.657 |
| ST * SD | 36.972 | 12 | 3.081 | 1.612 | 0.098 |
| C * ST * SD | 16.185 | 12 | 1.349 | 0.706 | 0.743 |
| Error | 214.067 | 112 | 1.911 | | |
| Total | 9,667.259 | 154 | | | |
| Corrected total | 1,227.797 | 153 | | | |

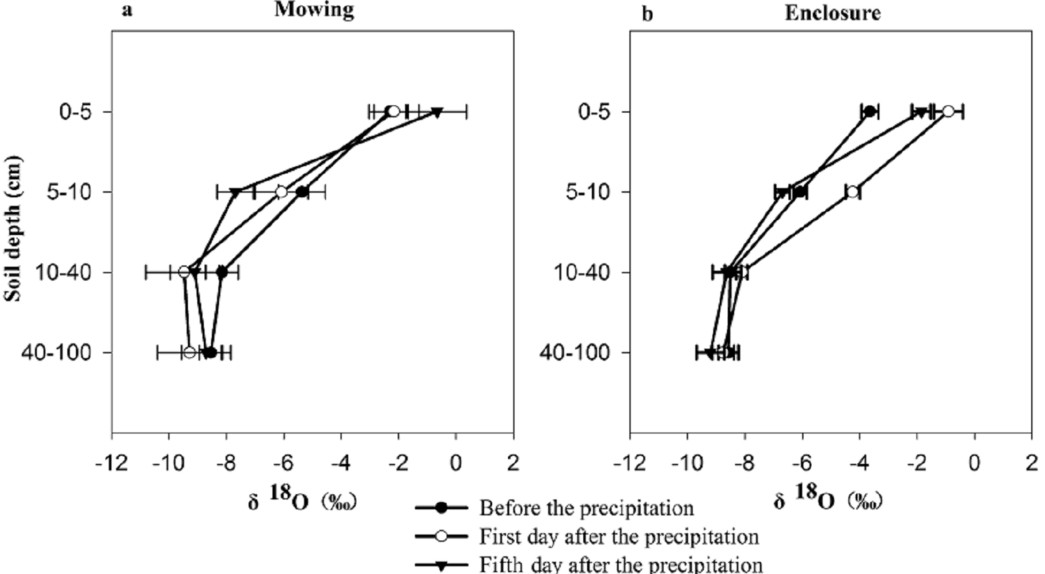

**Figure 2 $\delta^{18}O$ characteristics of soil water before and after precipitation under mowing (A) and enclosure (B) treatments.**

(Table 2). Compared with the values under the enclosure treatment, the water $\delta^{18}O$ for the deep layers under the mowing treatment showed a greater fluctuation with the increased sampling interval (Fig. 2). Under both treatments, with the SD increasing the $\delta^{18}O$ of soil water decreased gradually with a reduction from the surface layer to the middle layer of −5.85‰, −7.31‰ and −8.44‰, respectively, for the day before the precipitation and the first and fifth day after the precipitation under the mowing treatment, while a reduction of −4.86‰, −7.19‰ and −6.78‰ separately under the enclosed treatment for the three ST.

## Effects of mowing and summer precipitation event on the water $\delta^{18}O$ characteristics in plants

The mowing and STs had no extremely significant effects on the water $\delta^{18}O$ of any one of three plant species ($p > 0.05$), though the largest $\delta^{18}O$ value of the three plant species

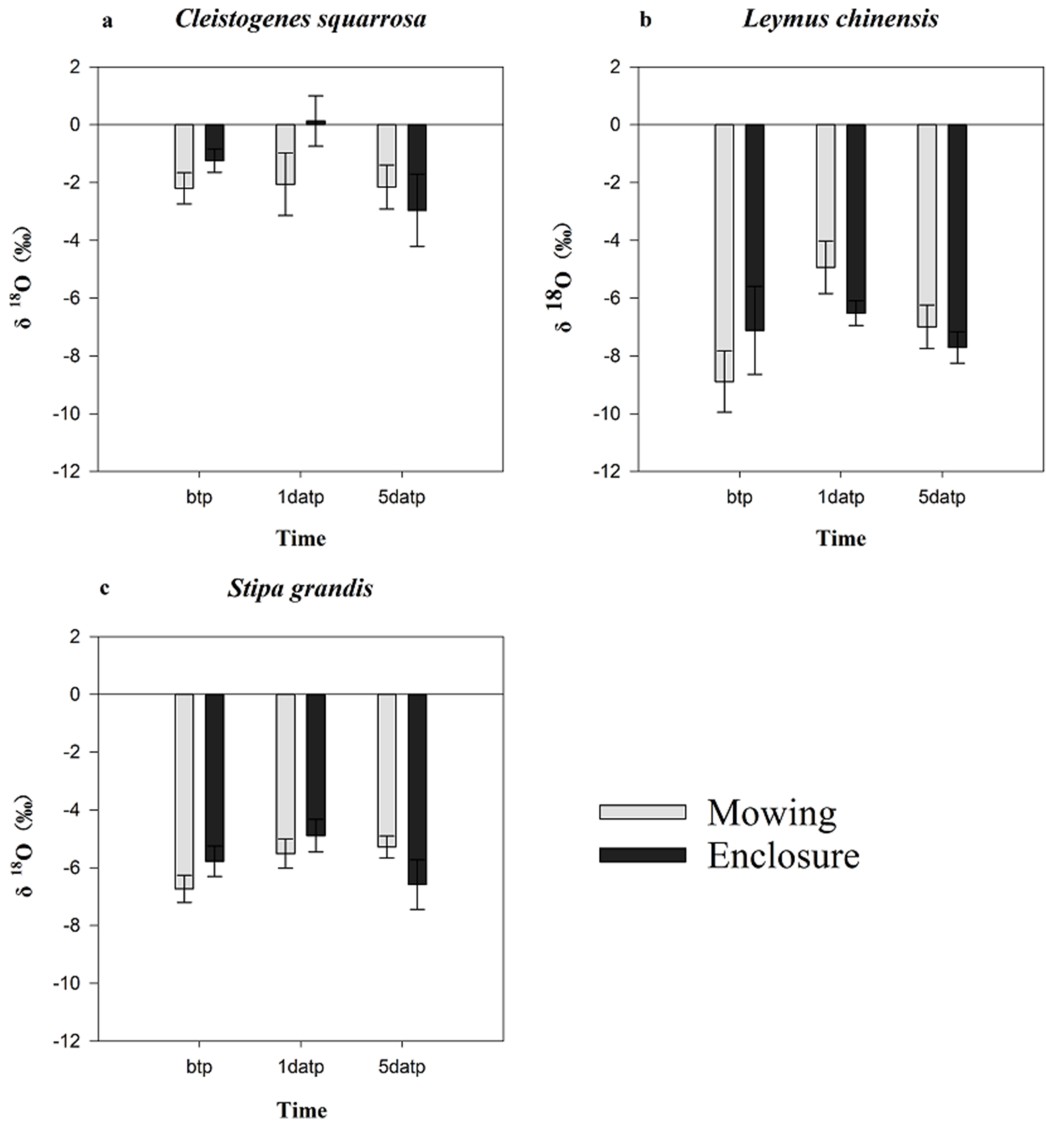

**Figure 3 Characteristics of the water δ¹⁸O in three plants before and after precipitation under mowing and enclosure treatments.** (A) Cleistogenes squarrosa, (B) Leymus chinensis, (C) Stipa grandis. bpt, before the precipitation; 1datp, the first day after precipitation; 5datp, the fifth day after precipitation.

appeared on the first day after precipitation (Fig. 3). Under the mowing treatment, the water $\delta^{18}O$ value of the three plant species ranged from $-8.891‰$ to $-4.934‰$ for *L. chinensis*, $-6.73‰$ to $-5.201‰$ for *S. grandis* and from $-2.202‰$ to $2.062‰$ for *Cleistogenes squarrosa*, with the minimum values of water $\delta^{18}O$ all occurring before the rainfall. Under the enclosure treatment, the minimum water $\delta^{18}O$ value of the three plant species all occurred on the fifth day after precipitation, and their water $\delta^{18}O$ ranged from $-7.709‰$ to $-6.523‰$ for *L. chinensis*, and $-6.584‰$ to $-4.889‰$ for *S. grandis* as well as $-2.964‰$ to $0.132‰$ for *Cleistogenes squarrosa*, respectively. The isotope values of plant xylem water were similar to those of soil water, indicating that the water in this soil layer is utilized.

**Table 3 Soil water utilization ratio of three plants in different soil layer under mowing and enclosure treatments before and after precipitation.**

| Sample date | Sample plot type | Plant species | The average contribution rate of each potential water source to plants (%) | | | |
|---|---|---|---|---|---|---|
| | | | 0–5 cm | 5–10 cm | 10–40 cm | 40–100 cm |
| Before precipitation | Mowing | *L. chinensis* | ~ | ~ | ~ | ### |
| | | *S. grandis* | 14 (0–20) | 25.5 (0–56) | 32 (0–66) | 28.5 (0–70) |
| | | *C. squarrosa* | ### | ~ | ~ | ~ |
| | Enclosure | *L. chinensis* | 14.2 (0–28) | 29.6 (2–58) | 22 (0–48) | 34.2 (20–48) |
| | | *S. grandis* | 36.5 (12–56) | 39 (0–88) | 20.9 (0–44) | 3.6 (0–8) |
| | | *C. squarrosa* | ### | ~ | ~ | ~ |
| The first day after precipitation | Mowing | *L. chinensis* | 50.8 (34–62) | 23.6 (0–60) | 14.4 (0–38) | 11.1 (0–30) |
| | | *S. grandis* | 39.9 (16–54) | 29.9 (0–82) | 15.5 (0–44) | 14.6 (0–46) |
| | | *C. squarrosa* | ### | ~ | ~ | ~ |
| | Enclosure | *L. chinensis* | 13.2 (0–28) | 21.7 (0–48) | 33.7 (0–78) | 31.4 (0–68) |
| | | *S. grandis* | 27.1 (0–48) | 35.9 (0–84) | 19.2 (0–50) | 17.8 (0–48) |
| | | *C. squarrosa* | ### | ~ | ~ | ~ |
| The fifth day after precipitation | Mowing | *L. chinensis* | 18.7 (10–24) | 30.2 (0–90) | 25.5 (0–70) | 25.6 (0–70) |
| | | *S. grandis* | 40.7 (36–44) | 21.4 (0–54) | 18.4 (0–48) | 19.4 (0–54) |
| | | *C. squarrosa* | 80.9 (80–82) | 7.1 (0–14) | 7.7 (0–16) | 4.3 (0–10) |
| | Enclosure | *L. chinensis* | 9 (0–20) | 25.8 (0–58) | 33.2 (0–82) | 32 (0–76) |
| | | *S. grandis* | 22.6 (4–34) | 32.8 (0–92) | 23 (0–64) | 21.5 (0–60) |
| | | *C. squarrosa* | 81.6 (78–84) | 8.4 (0–20) | 5.1 (0–12) | 4.9 (0–12) |

Notes:
The symbol "###" refers to the soil layer where major water source is from for the examined species, as the $\delta^{18}O$ value of plant stem water is higher than the soil water in top soil layer or lower than bottom soil layer.
The symbol "~" refers to the soil layer where no water source is from for the examined species.

## Effects of mowing and summer precipitation event on the water sources of the three dominant plant species

Mowing and precipitation events changed the water sources of the plants (Table 3; Fig. 4). As the estimated range of the proportion that a plant species took up from various soil layers are relatively wide and overlapping, we could not quantify the percentage of water taken up from a specific soil layer. A general pattern in the results (Table 3) was that *Cleistogenes squarrosa* took up water majorly from top soil layer, *L. chinensis* took up relative more water from deep soil layer, and *S. grandis* took up water from the middle layer or from across various layers (Fig.4). Also, For *L. chinensis* and *S. grandis* in mowing grassland, the contribution rate of water form the upper soil layer to the plant was separately 50.8% and 39.9% on the first day after precipitation, which was higer than those of other soil layers. However, on the fifth day after precipitation, *L. chinensis* tended to be stabilized to take up water from the shallow to deep soil layers with the ratio ranged from 25.5% for middle soil layer and 30.2% for shallow soil layer, while *S. grandis* still took up more water (40.7%) from the upper soil layer than other soil layers. In contrast, these two species in enclosed grassland showed no sensitive change before and after the precipitation. *Cleistogenes squarrosa* used the

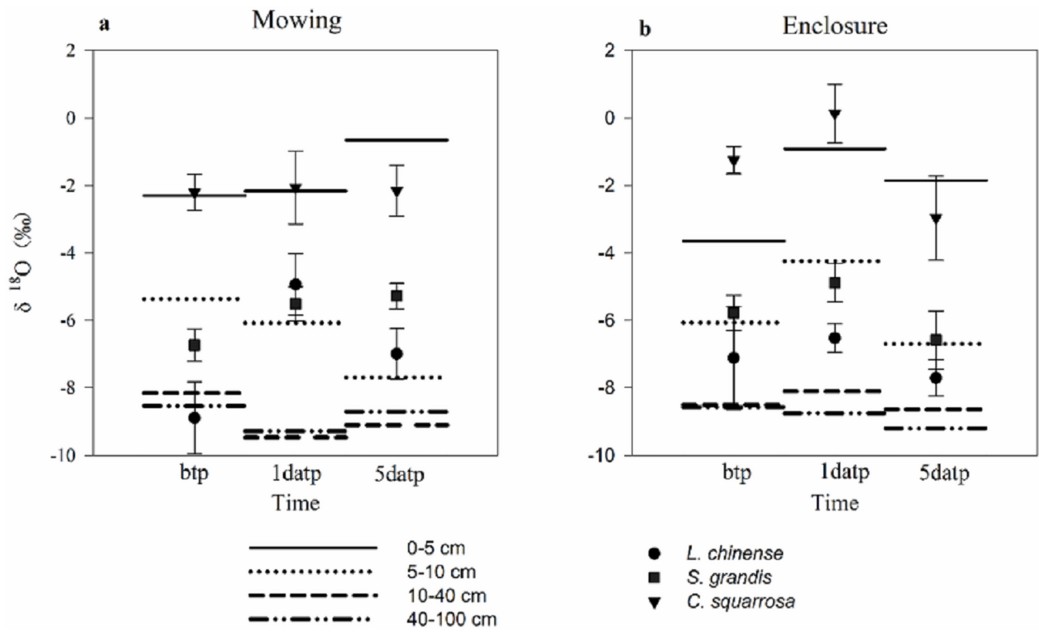

**Figure 4 Soil water source of three plants under mowing (A) and enclosure (B) treatments before and after precipitation.** bpt, before the precipitation; 1datp, the first day after precipitation; 5datp, the fifth day after precipitation.

water majorly from the top soil layer either under mowing or in enclosure, and either before and after precipitation events.

## DISCUSSION

Soil water comes primarily from precipitation, and the precipitation amount and pattern drives soil water dynamics (*Wang et al., 2013*). Combining the characteristics of soil water isotopes with the analysis of soil water content can provide information on the soils role in the migration of rainwater. Our results show that the soil water content in the 0–40 cm soil layer has similar difference between mowed and enclosed grasslands, either before or after precipitation, which may be related to vegetation cover. Mowing grassland has low vegetation and litter cover, that is, a high degree of soil exposure, thus may lead to a high evaporation of water from the top soil layer. By contrast, the enclosed grassland has high vegetation and litter cover, and thus the evaporation through soil surface is reduced (*Ma, Wang & Li, 2009*). Our results also show that with the increasing SD, the soil water content under the mowing treatment gradually increased relative to that of the enclosure treatment. This might be related with dual effects of mowing on water processes in soil profile: the first was a facilitation to water infiltration during the precipitation event because of no litter coverage (*Wang, 2011b*); and second was a reduction in water uptake from deep soil layer due to less plant transpiration.

Evaporation would result in the enrichment of δ$^{18}$O in the surface soil water. Our results show that with increasing SD, the evaporation decreases, and the isotopic abundance decreases approximately exponentially with the depth until it reaches a relatively constant abundance, consistent with the results of *Xu et al. (2012)*. There was no significant
difference in the precipitation infiltration or the fractionation degree of evaporation processes between the mowing and enclosure treatments, but the $\delta^{18}$O in the soil water at different SDs under the two treatments were both significantly affected by precipitation. After precipitation, the soil was still affected by intense evaporation. In our study, in shallower soil, the heavy isotope in the soil was enriched and the $\delta^{18}$O increased, which agreed with previous studies (*Tian, Yao & Sun, 2002*; *Jin et al., 2015*; *Wang et al., 2009*). Previous studies have shown that *L. chinensis* and *S. grandis* respond differently to changes in the precipitation amount and pattern. Studies also show that mowing reduces the density, height and biomass of *L. chinensis* (*Guo, 2017*). In this study, the three dominant plant species *L. chinensis*, *S. grandis* and *Cleistogenes squarrosa* exhibited a large difference in water source before and after precipitation. Specifically, *L. chinensis* in mowing grassland used the water majorly from the deep soil layer (40–100 cm) in dry soil before precipitation, but substantial proportion of water uptake was from top soil layer on the first day after precipitation, and the proportion declined gradually following the precipitation; while *L. chinensis* in enclosed grassland took up water from various soil layers, did not exhibit a large fluctuation. Similarly, *S. grandis* took up proportionally more water from the top soil layer after than before precipitation in mowing grassland, while it showed no much difference in water source before and after precipitation in enclosed grassland. Mowing had no significant effect on the water source of *Cleistogenes squarrosa*, which utilized soil water majorly from the top soil layer in all cases. Our findings that *L. chinensis* tends to use more water from deep soil layer in comparison to *Cleistogenes squarrosa* and *S. grandis*, is supported by the findings of *Yang et al. (2011)*. These results also indicate that *L. chinensis* and *S. grandis* in mowing grassland are subject to more severe water deficit in top soil layers before the precipitation (Fig. 1), thus more sensitive to precipitation events than that from the enclosed natural grassland. However, not just rainfall intensity, but the land topography (e.g. slope) also impact the soil water distribution and thus the plant water sources. In our study, we only studied the water source of the three species before and after the light and medium rain events, the signatures of which may not adequately describe soil zones explored for water uptake.

The utilization of soil water by plants is closely related to the distribution of plant roots (*Xu & Li, 2006*). *Leymus chinensis* is a rhizome grass, and its rhizomes are mainly distributed in the 5–10 cm soil layer, while the roots are mainly in the 0–30 cm soil layer; *S. grandis* is a tall bunchgrass, and its roots are also concentrated in the 0–30 cm soil layer. *Cleistogenes squarrosa* is a short bunchgrass, with root system concentrated in the 0–10 cm soil layer (*Chen, 2001*; *Ma, 1989*; *Zhu, 2004*). These differences in root morphology and distribution pattern among these three species may partly explain the observed differences in their water uptake from different soil layers. The difference between the *L. chinensis* and *S. grandis* root systems warrants further studies to confirm the observed fact that *L. chinensis* took up proportionally more water than *S. grandis* from deep soil layers.

## CONCLUSION

In this study, we explored the water use sources of three coexisting dominant plant species before and after precipitation events including *L. chinensis, S. grandis* and *Cleistogenes*

*squarrosa* growing in both enclosed and mowing grassland in a typical steppe. We found that the soil moisture and its $\delta^{18}O$ were not affected by mowing, and only the soil moisture changed significantly after the precipitation. The three dominant plants showed divergent water sources with *Cleistogenes squarrosa* generally taking up water from the top soil layer (0–10 cm), *L. chinensis* taking up relative more water from deep soil layer (>70 cm), and *S. grandis* just taking up water from the middle to deep soil layers (10–70 cm). In addition, *L. chinensis* and *S. grandis* in mowing grassland tended to take up more water from the upper soil layers following precipitation events relative to those in the enclosed grassland, indicating a more sensitive change of soil water sources for the two species in mowing than enclosed grassland use. Our results have important theoretical values for understanding the water competition among plants in fluctuating environments and under different land use in the typical steppe.

### Funding

This work was supported by the National Key Research and Development Program of China (2016YFC0500503), the Technical System Project of the National Herbage Industry, China (CARS-34) and the Major Program of the Natural Scientific Foundation of Inner Mongolia Autonomous Region, China (2014ZD02), and Science and Technology Project of Inner Mongolia, China (201502098). The funders had no role in study design, data collection and analysis, decision to publish, or preparation of the manuscript.

### Grant Disclosures

The following grant information was disclosed by the authors:
National Key Research and Development Program of China: 2016YFC0500503.
Technical System Project of the National Herbage Industry, China: CARS-34.
Natural Scientific Foundation of Inner Mongolia Autonomous Region, China: 2014ZD02.
Science and Technology Project of Inner Mongolia, China: 201502098.

### Competing Interests

The authors declare that they have no competing interests.

### Author Contributions

- Tiejun Bao conceived and designed the experiments, analyzed the data, prepared figures and/or tables, authored or reviewed drafts of the paper.
- Yunnuan Zheng performed the experiments, prepared figures and/or tables.
- Ze Zhang performed the experiments.
- Heyang Sun performed the experiments.
- Ran Chao performed the experiments.
- Liqing Zhao contributed reagents/materials/analysis tools.
- Hua Qing conceived and designed the experiments.
- Jie Yang contributed reagents/materials/analysis tools, approved the final draft.

- Frank Yonghong Li analyzed the data, authored or reviewed drafts of the paper, approved the final draft.

## Data Availability

The raw measurements are available in the Supplemental File.

## Supplemental Information

Supplemental information for this article can be found online at http://dx.doi.org/10.7717/peerj.7737#supplemental-information.

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
