# Peer review of "Divergent water sources of three dominant plant species following precipitation events in enclosed and mowing grassland steppes"

_PeerJ, doi:10.7717/peerj.7737_

## Round 0.1 · original submission · Major Revisions

This paper addresses an important topic of plant water sources in a semi-arid climatic setting. The methods and experimental treatment appear to be sound. However, reviewers #1 and #3 have raised concerns about the clarity of the writing. Each has offered some detailed suggestions for improving the writing and expression. The reviewers have also raised concerns about the proper use of significance, drawing a clear distinction when statistically significant results are being discussed or not. The discussion section in particular needs improvement in this regard. Reviewer #2 suggests the paper would be improved by inclusion of a conclusion section. Please read these reviews carefully and provide detailed responses as you revise the manuscript.

Reviewer 1 ·

Basic reporting

General Comments
The study entitled, ‘Divergent water sources of three dominant plants at precipitation events in natural and mowing grassland of a typical steppe’ attempted to determine water sources of three dominant plant species growing in mowed and natural steppes. The method of cryogenic vacuum extraction of water, which was used in the study, is one of the most widely used and accurate extraction methods to obtain plant and soil water samples for δ18O. This task is daunting, but produces valuable data. Thus, the method used is very appropriate.

The study has great potential, especially in informing restoration ecologists on possible choices of plant species for different niches. However, the handling of the data and its analysis could have been slightly different. I have now highlighted these suggestions in the results section of this report and also as comments on the attached PDF version of the manuscript.

Language: The manuscript has several grammatical mistakes that need attention in order to improve on clarity. I have highlighted some of these in the manuscript.
Title: Firstly, I suggest revision of the title to ‘Divergent water sources of three dominant plant species following precipitation events in natural and mowed grassland steppes’.
Abstract: see corrected pdf text for many specific comments
Introduction: I suggest removal of the first sentence (line 36-37) as it is not adding value to the manuscript.
Line 62-63 Provide a more accurate description of the study sites. This is not an accurate description of the ecosystems used in the study. 'coastal' not 'riverbanks' Mediterranean-type deserts' not ‘temperate steppes'
Line 64 Its not true, many studies done in South Africa are in semi-arid areas
Line 81 The sentence is too long and wordy. Consider rephrasing it
Methods:
Line 101 What is the relevance of accumulated temperature? It gives an impression of heat storage.
Line 106 You have only introduced, 1. Natural grassland and 2. Mowing grassland. ‘Experimental grassland’ will confuse readers. Rather refer to experimental area
Line 114 Rephrase for clarity. Do you mean that: Two precipitation events were recorded in July 2016, which are 10.8mm on the 29th of July and 20.0mm on the 30th of July.
Line 114 Rephrase to ‘Experimental data were analyzed using SPSS Version 19.0 (SPSS Inc., Chicago, USA)’
Line 153 rephrase to ‘which are 1) surface water layer (0-5cm), 2) shallow water layer (5-10), 3) middle layer (10-40) and 4) deep water layer.’
Line 154 Change ‘The’ to ‘An’
Line 155 Provide the source of the model. Include in parenthesis the source. 
Results:
Revise figures as follows:
Figs 1 & 2 Because four depth categories were mentioned in line 153 (0-5; 5-10; 10-40, >40) , this figure should be redrawn to show average moisture values of these categories, possibly on a secondary axis or as a different fig with the categories plotted against δ18O value and moisture contents. Then relate species δ18O values to soil layer (0-5; 5-10; 10-40, >40) averages. Then conclusions around soil layer would make more meaning to the readers
Figure 3 Refer to ‘three plant species’ not ‘3 plants’. It implies that you used three individual plants, one of each species.
Figure 4 Symbols are missing. This is making your results difficult to understand. eg. What do different lines stand for? What does a circle or a triangle mean?
Line 162 This is difficult to understand. The authors mention a ‘very significant change’ but ‘no significant effect was detected’? What 'effect' was measured here?
Line 165 This is poorly constructed. You mention that 'changes were different' but we know that a change is a difference between two states/levels etc. Please rephrase
Line 167-169 This section needs careful description. I suggest making reference to the four categories that were identified earlier in line 153. What was the purpose of separating soil zones if interpretation is made based on all individual distances measured.
Line 183-184 I think one cannot use ‘largest δ18O value’ if there were no significant differences observed in δ18O values. Please rephrase to correctly to relay the intended message.
Line 186 Delete the word ‘separately’
Line 202-208 Indicate the proportions used in this section.
Discussion
I suggest that you consider revising the current discussion of your results after considering δ18O values for each layer 0-5; 5-10; 10-40 and >40. These δ18O values for soil layers are the potential water sources that should be present in the non-photosynthetic parts of the plant. I think this is still possible with the Isosource model. Signatures of precipitation events (2 rain events) may not adequately describe soil zones explored for water uptake.
Line 214 I suggest that you rephrase this. It suggests lack of confidence in your statistical results. Should be ' were no significant differences between mowed and enclosed grasslands’.
Line 216 Pay attention to your grammar. Use past tense in this section. Rephrase. I could not follow the statement here. Same applies to line 222-225
Line 266 How is shown from your data?

Experimental design

see previous comments in pdf version

Validity of the findings

A revision is suggested after considering suggested comments

Additional comments

see previous comments

Annotated reviews are not available for download in order to protect the identity of reviewers who chose to remain anonymous.

Reviewer 2 ·

Basic reporting

Line 19, delete extra space;

Line 28, "distribution pattern" is vague, precipitation distribution?;

Line 37, this paragraph of "Utilization of different water sources by plants.." is not very relevant to this study, as this study is not actually characterizing the water sources but only the depth of soil layer for water uptake.

Line 51, this example, "In the Sequoia sempervirens forest.." is very extreme because those trees are very tall and at high elevation. And is not appropriate as an example for this study because plants involving in this study are perennial grasses and shrubs.

Line 61-64, again, authors should focus on expanding the Introduction discussing the gaps in those few studies with steppe ecosystems. Mentioning temporal forests is not useful for this study because they are different ecosystems.

Change captions for table 1 and 2 to "ANOVA tables for ....."

Experimental design

Line 99, add "China"

Line 107, "The experiment was started..";

Line 112, "of each year"? how long was mowing period or only in 2011

Line 122, ''..and combined as one replicate"

Validity of the findings

Line 162-163, I am confused by this sentence. soil water content had a significant change but then no significant effects were detected? so the change before and after precipitation is not significant?

Line 167, should be more specific, at what depth the soil water content started to decrease. Same here in line 169, at what depth there is a reverse?

Line 178, it might be better to calculate % change instead of saying a rapid decrease and a mild decrease.

Line 187-193, I think the results for those isotope should not just be values because they are hard to understand. Authors should give some topic sentences say what those ranges mean comparing among three species.

Line 214, if the effect is not significant, either there is no sufficient data or more variables (factors) should be considered during the experiment. Authors should not explain the results assuming it is significant. Again, in line 229-230, if the result is not significant, why still have implications? what kind of implications? or authors should pick the experimental site with longer mowing history.

Line 235, were there any recent references to be compared with? this paper (and is the only reference here) was from 1983.

Line 240, "In our study, in shallower soil, ...."

Line 244, Since authors discussed the time intervals between the two collections (first date vs. fifth date) below, the example of reference should also be the intervals between sample collections instead of the time intervals between precipitation events.

Line 260, omit "(Yang et al., 2011)"

Need a conclusion section? if this is required by the journal.

Additional comments

This paper reported changes in soil moisture at different depths in mowing vs. enclosed grassland before and after precipitation events. It also studied the oxygen stable isotope ratios of soil water and stem water of three plant species: Leymus chinensis, Stipa grandis and Cleistogenes squarrosa. This paper over discussed some results for which the statistical analyses were actually not significant. Introduction and discussion need more work, please see the above comments.

Reviewer 3 ·

Basic reporting

This manuscript aims to explore how the three coexisting dominant species Stipa grandis, Leymus chinensis and Cleistogenes squarrosa in a typical steppe community respond to summer precipitation event in water use under two grassland utilization modes (mowing and enclosure). I think that the manuscript addresses interesting questions. However, the manuscript was not well written. There are too many grammatical errors, and the manuscript should be improved greatly in English expression.

I have some important concerns and suggestions about this study.

The abstract should be rewritten because of some grammatical errors.
Line 24 “S. grandis from ……..” should be “S. grandis took up water from ……..”
Line 27 “a more sensitive soil moisture change” should be “a more sensitive change of soil moisture”
Line 29 “the observed differences” should be “the differences”
Line 30-31 “Plant root systems……..”. This sentence doesn't make sense.

You should give a better background of oxygen stable isotope ratios (δ18O) of soil water in introduction section. Why you choose oxygen stable isotope ratios (δ18O) of soil water to explain divergent water sources of three dominant plants, and why you choose Stipa grandis, Leymus chinensis and C. squarrosa as research objects, please increase the explanation and provide references.
Line 60 “the studies on” should be “studies about”
Line 61 “the ecosystems of” can be deleted.
Line 63 “the coasts” should be “coasts”
Line 68 “the semi-arid steppe ecosystem” should be “semi-arid steppe ecosystems”.
Line 69 “the soil water” should be “soil water”
I think there are too many “the” in this manuscript.
Line 79 “changes in” should be “changes of”
Line 81-84 “Therefore……..” should be “Therefore, understanding the mechanisms of plant water use is essential to steppe management by exploring how dominant plants respond to instantaneous precipitation events and use available water resources in natural and mowing conditions .
Line 87 “event” should be “events”
Line 2,18,28,88 The expression about utilization modes should be consistent. “mowing and natural” in the title, “native (in enclosure) and mowing” and “mowing and enclosed grassland” in the abstract, “mowing and enclosure” in introduction section.
Line 88 “did this by detecting” should be “determined”
Line 90-91 “mowing zone and enclosure zone of the steppe” should be “mowing and enclosure plots”
Line 92-94 English language

The methods section needs a lot of improvements, because there are many aspects that are missing or unclear.
Line 100 “a mean annual temperature of -0.4 °C”. Please check the data “-0.4 °C” and provide the year range of calculating the mean annual temperature. I think a mean annual temperature in this site should be about 2°C .
Line 101 “the average plant growth period” should be “the average period of plant growth”
Line 102 “around 200 mm ~ 350 mm” should be “between 200 to 350mm”
Line 105 “or Calcic-orthic Aridisol” should be “equivalent to Calcic-orthic Aridsoils”
Line 106 “The experimental grassland was” should be “The vegetations in the study region are ”
Line 107-109 “The experiment……..” should be “”The enclosure and mowing plots were set up in 2011 to study the effects of annual mowing on native steppe.”
Line 109-111 “A total……..” should be “eight 20 m×30 experimental plots (four enclosure and four mowing plots) were established with a distance 2m between any two plots.”
Line 114-116 “Two precipitation……..” should be “a light precipitation event of 10.8 mm on 29 July and a medium precipitation event of 20.0 mm on 30 July.
Line 118-119 “Plant samples ……..” should be “Non-photosynthetic tissues of plant from the interface”

In results section, you should reanalyze your data.
Line 161 “Effects of ……..” should be “Effects of summer precipitation on soil water content in mowing and enclosure steppe”
Line 163 “but no significant effect was detected (Table 1, Fig. 1)”, I think there were significant effects of summer precipitation on soil water content of 5-10 cm and 10-20cm in mowing and 5-10 cm in enclosure steppe. Soil water content after precipitation was higher than before precipitation.
Line 170-171 “Effects of ……..” should be “Effects of summer precipitation onδ18O of soil water in mowing and enclosure steppe”. You should focus on the differences between two plots. There were significant differences inδ18O of soil water 0-5 cm and 5-10cm in enclosure steppe before and after precipitation events.

There are some inappropriate expressions in Note of Table 3. It is easy to make the reader confused. You should explain “~” in Note of Table 3.
There was no legend in Fig. 4.

References
The references should be in the same format.

Overall, I think this MS should be revised to reach the standard of publication in this journal.

Experimental design

no comment

Validity of the findings

no comment

---

## Round 0.2 · Minor Revisions

There are just a few additional minor changes suggested by a reviewer. Please include p value as suggested since this provides the reader with additional potentially important information even if the value is <0.05

Reviewer 2 ·

Basic reporting

None

Experimental design

Line 101, why there is a > 0 degree C next to the accumulated temperature?

Validity of the findings

Line 159, subheading can just be "Results", remove "and analysis"

Line 185, i again vote to have the actual value even if it is larger than 0.05. In ecology, sometime if p values fall into 0.05 and 0.1, we might also see that as a weak significant differences. Because sometimes researchers not seeing a significant difference does not mean there is not, it might be the small sample size or larger variation.

Line 220, change to "were similar between..."

Line 262, not just rainfall intensity, but the land topography (e.g. slope) also impact the soil water distribution.

Line 281-285, this is the conclusion part, so I suggested authors to add the details of how they define "middle, deep soil layer" by giving the range of depths.

Additional comments

I appreciated authors' efforts to revise this paper. I think this paper can be accepted after some minor edits suggested above.

---

## Round 0.3 · accepted · Accept

I have examined the authors' responses to the most recent review of the manuscript and am satisfied with the responses and implemented changes. I am happy to accept this paper for publication.